# Multi-Fidelity Physics-Informed Neural Networks (PINN) with Boundary-Aware Losses for Ice-Bed Topography Prediction

## Abstract

Predicting ice dynamics and sea-level rise requires an understanding of subglacial bedrock topography; however, inversion remains a challenging task in data-sparse regions where surface observations are limited. Some conventional machine learning methods face challenges in predicting subglacial topography due to heavy reliance on purely data correlations and cannot guarantee physical consistency, especially in data-sparse regions. Physics-Informed Neural Networks (PINNs) address this limitation by embedding partial differential equation (PDE) constraints into deep learning, enabling more physically consistent predictions. However, most existing PINN formulations depend on a single fidelity of physics, and soft boundary penalties can still compromise performance. We propose a multi-fidelity PINN framework for ice-bed topography prediction that advances beyond these limitations in two ways. First, we introduce multi-fidelity residual coupling, jointly enforcing the shallow-ice approximation (SIA) and reduced-Stokes equations within a single network. This coupling improves accuracy while maintaining physics consistency, achieving strong predictive performance (e.g., Test MSE = 0.028, and $R^2$ = 0.97). Second, we design a boundary-aware weak-form loss that supports traction/flux (Neumann) and optional Dirichlet constraints, allowing flexible enforcement of margin physics. Experiments show that hard Dirichlet enforcement over-constrains the model and reduces accuracy, while soft or selective enforcement preserves predictive quality. To our knowledge, this is the first Physics-Informed Neural Network (PINN) framework for predicting ice-bed topography that unifies multi-fidelity partial differential equation (PDE) residuals with configurable boundary-aware losses, providing a practical and extensible approach to physically plausible predictions in data-sparse regimes.

## 1 Introduction

Understanding and predicting the dynamics of ice sheets is central to understanding future sea-level rise, one of the most pressing global challenges of climate change. A key in this effort lies in presuming the subglacial bedrock topography, which strongly governs ice flow but is poorly observed due to the inaccessibility of the ice–bed boundary. While airborne radar and seismic surveys provide direct measurements, coverage is sparse, particularly in fast-flowing outlet glaciers where uncertainties in bed geometry lead to large uncertainties in dynamical projections Morlighem et al. (2017). As a result, the development of reliable inversion techniques for bed topography remains an open and important research problem in glaciology. Recent advances in machine learning have offered opportunities to tackle this challenge by learning statistical relationships between surface observations (e.g., elevation, velocity, mass balance) and basal conditions Xiang et al. (2022); Krishna et al. (2023). However, purely data-driven models are prone to overfitting and can fail to generalize in data-sparse regions, where extrapolation requires strong physical priors. More recently, Physics-Informed Neural Networks (PINNs) have gained attention as a framework for embedding physical laws into learning Karniadakis et al. (2021); Raissi et al. (2019). Penalizing violations of the governing partial differential equations (PDEs), PINNs enforce physics consistency while reducing dependence on labeled data. Yet, most PINN formulations in the earth sciences remain limited in two key respects. First, existing PINNs often adopt a single fidelity of physics, such as the shallow-ice

approximation (SIA) or a reduced-Stokes model. This discards complementary strengths: SIA is computationally efficient but oversimplified in complex flow regimes, while Stokes captures higher-order dynamics but at a higher cost. Second, boundary conditions are usually handled via soft penalty terms, which can either under-constrain the model and, when applied too strongly, degrade accuracy near glacier margins. In this paper, we present a multi-fidelity, boundary-aware Physics-Informed Neural Network (PINN) framework for predicting ice-bed topography. Our contributions are threefold: Multi-fidelity residual coupling. We jointly enforce the shallow-ice approximation (SIA) and reduced-Stokes equations within a single network. This coupling integrates complementary physics across fidelities, improving predictive accuracy while maintaining physical consistency.

Boundary-aware weak-form loss. We design a flexible boundary formulation that incorporates both traction/flux (Neumann) and optional Dirichlet constraints. This allows selective enforcement of margin physics, avoiding the over-constraint associated with hard Dirichlet penalties.

Empirical validation. We evaluate our method on glacier datasets, showing strong predictive skill (Test MSE = 0.028, $R^2$ = 0.97) while tightly satisfying PDE constraints. Results demonstrate that hard Dirichlet enforcement reduces accuracy, whereas boundary-aware enforcement preserves predictive quality.

To our knowledge, this is the first PINN framework for ice-bed inversion that unifies multi-fidelity PDE residuals with configurable boundary-aware losses. By combining physically consistent learning with flexibility at domain margins, our approach provides a practical and extensible pathway toward more reliable ice-sheet models, with direct implications for projecting sea-level rise in data-sparse regimes.

## 2 LITERATURE REVIEW/RELATED WORK

Conventional Machine learning has been explored for subglacial topography Yi et al. (2023) systematically evaluated machine learning and statistical models (e.g.,gaussian process regression, XGboost, dense neural network, long-short term memory, variational auto-encoder etc.) for Greenland bed prediction, showing promise but highlighting limits in extrapolation and physical consistency. Traditional mass-conservation techniques Morlighem et al. (2011) and datasets including BedMachine v5 NSIDC (2023) GIMP DEM Howat et al. (2014), and MEaSUREs velocity mosaics continue to serve as the foundation for inversion techniques. Classical adjoint-based inversions and Bayesian UQ Petra et al. (2014); Isaac et al. (2015) provide gold-standard benchmarks but are computationally costly. Physics-informed ML complements these approaches by offering scalable, data-efficient, and physically grounded learning. Physics-informed learning addresses this by embedding PDE structure. For example, Jouvet & Cordonnier (2023) proposed a PINN-based ice-flow emulator, while Cheng et al. (2024) applied PINNs directly for forward and inverse ice-sheet modeling at Helheim Glacier. These works prove feasibility but typically adopt a single fidelity of physics and uniform boundary treatment. Hybrid neural-operator / FEM methods further indicate that learned operators can replace costly inner solves while honoring PDE constraints He et al. (2023). Likewise these works typically fix a single fidelity and use uniform boundary penalties. Physics-Informed Neural Networks (PINNs) embed governing PDEs and boundary conditions in the training objective to solve forward and inverse problems with improved physical consistency Raissi et al. (2019). However, PINN training can be fragile, motivating domain decomposition Jagtap et al. (2020) and weak-form variants Kharazmi et al. (2021) to stabilize residual enforcement. A recent large-scale evaluation, PINNacle Hao et al. (2024), provides a systematic benchmark across PDE classes and highlights open challenges in stability, loss balancing, and boundary enforcement—challenges directly motivating our boundary-aware weak form. However, our approach differs by unifying multi-fidelity SIA + Stokes and boundary-aware weak-form losses.

Multi-fidelity learning and residual weighting: Learning from multiple fidelities improves data efficiency and accuracy Meng & Karniadakis (2020). This study shows that incorporating multiple sources of data improves robustness. Additionally, composite networks leverage multi-fidelity inputs, and uncertainty-based weighting Kendall et al. (2018) balances competing objectives adaptively. It provides a principled way to adapt residual coupling without brittle manual tuning—supporting our multi-fidelity residual coupling claim where SIA and reduced-Stokes are enforced jointly with learned weighting. our method differs by coupling SIA + reduced-Stokes in one network and by selective boundary enforcement. Boundary conditions in PINNs: Enforcing bound-

ary conditions has long been a bottleneck. Handling nonhomogeneous boundary conditions within PINNs has been studied Dwivedi et al. (2020), while hard-constrained methods reduce penalty tuning but risk the over-constraining solutions. Many implementations still rely on uniform soft penalties, which can over- or under-constrain margins, precisely what our boundary-aware weak-form addresses. In glaciological settings, where marine termini and flux boundaries are critical, uniform soft penalties can bias inversion. Our boundary-aware weak-form builds on this literature by allowing selective Neumann/Dirichlet enforcement.

## 3 DATASET BACKGROUND AND PREPOSSESSING

The dataset used in this study consists of 632,706 data points derived from radar-based bed elevation measurements from the Upernavik glacier system in West Greenland. These data points of radar-derived bed elevation measurements are target values, which are track_bed_x and track_bed_y (coordinates of radar data points (m)) and track_bed_target: subglacial bed elevation along the flight line. Four key sources: Surface Elevation data from the Greenland Ice Mapping Project (GIMP), ice flow surface velocity data in both longitudinal and latitudinal directions from satellite interferometry datasets (Landsat-8, Sentinel-1, RADARSAT-2), ice thinning rates from ICESat-2, and surface mass balance from the Regional Atmospheric Climate Model (RACMO). These data points have ground truth that is used to measure the accuracy of the predictive models. Since these data points are only along the tracks of the airborne radar sensor, they are referred to as "Track Data." Yi et al. (2023). Since the Track Data themselves only provide bed elevations along flight lines without additional surface descriptors, these external datasets were combined and co-registered to form an integrated dataset of 630k training samples. The combined dataset was cleaned by removing unnecessary columns; we standardized the data and split it into 80% training and 20% testing to maintain model evaluation integrity, ensuring that training data points were not too similar to those in the testing set. The final feature set used in model training consisted of surface velocity (surf_vx, surf_vy), surface elevation (surf_elv), ice thinning rate (surf_dhdt), and surface mass balance (surf_SMB). These steps streamlined the dataset for subsequent analysis and physics-informed model training. We use the standardized *trackbed* dataset paired with radar-derived bed elevations. In the baseline and multi-fidelity settings, the features are scaled with `StandardScaler`, and the target is optionally normalized. For the boundary-aware ablation, we additionally construct Dirichlet boundary labels by applying nearest-neighbor interpolation of radar-derived bed elevations along glacier margins. These interpolated labels are then combined with Neumann flux conditions to form the boundary-aware weak-form loss.

## 4 EVALUATION METRICS

This work focuses on ice-bed topography prediction, using a multi-fidelity PINN framework that leverages boundary-aware losses for more accurate PDE-constrained learning tasks, evaluated with both standard regression metrics and physics-informed loss components. The regression metrics include Mean Absolute Error (MAE), Root Mean Squared Error (RMSE), Mean Squared Error (MSE), and the coefficient of determination ($R^2$). MAE provides an intuitive measure of prediction accuracy, RMSE emphasizes large deviations by penalizing larger errors, and $R^2$ quantifies how well the model explains variance in the data (values closer to 1 indicate better fit). MSE is used both as a training loss term and as a performance metric. In addition to predictive accuracy, physical consistency is enforced through a physics-informed loss that embeds glaciological partial differential equations (PDEs). Specifically, the framework combines the data-driven MSE with multi-fidelity PDE residuals, jointly enforcing the shallow-ice approximation (SIA) and a reduced-Stokes formulation. These residuals quantify how well the neural network satisfies governing momentum-balance equations, improving physical plausibility in data-sparse regions. A boundary-aware weak-form loss supplements this by enforcing Neumann flux (traction) conditions and, in ablation tests, selectively applying Dirichlet constraints inferred from observed bed topography. Additionally, we report the mean-squared residuals of the SIA and Stokes equations, as well as boundary weak-form terms, to explicitly quantify physics satisfaction alongside data-driven accuracy. Combining both regression metrics and residuals based on physics, the proposed framework achieves a balance between predictive accuracy and adherence to physical laws, allowing PINN to provide robust and physically consistent subglacial topography predictions Raissi et al. (2019).

## 5 METHODOLOGY

Our approach is a multi-fidelity Physics-Informed Neural Network (PINN) framework for inferring subglacial bedrock topography from surface observables, which integrates (i) multi-fidelity residual coupling of the shallow-ice approximation (SIA) and reduced-Stokes equations and (ii) a boundary-aware weak-form loss that flexibly enforces traction/flux (Neumann) and optional Dirichlet conditions. Figure 1 provides a schematic of the overall pipeline of the multi-fidelity plus boundary aware PINN workflow.

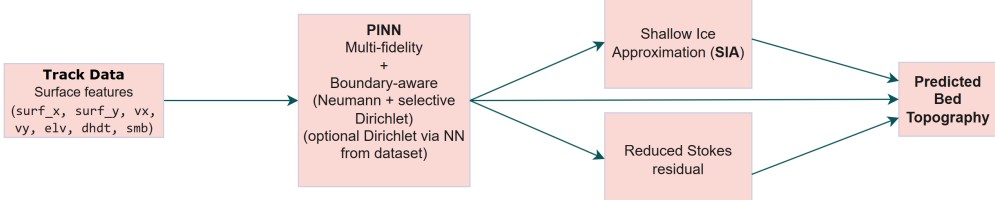

Figure 1: Training flow: Multi-fidelity + Boundary-aware PINN

### 5.1 MODEL ARCHITECTURE

This project (Multi-fidelity + boundary-aware PINN) uses a fully connected multilayer perceptron with three hidden layers (64–128 neurons, tanh activations). Parameters are initialized using the Xavier initialization Glorot & Bengio (2010). The network maps scaled surface features.

$$x \mapsto \hat{b}(x), \quad \text{where } \hat{b}(x) \text{ represent the predicted subglacial bed topography}$$

Two key modules extend the standard PINN formulation Raissi et al. (2019).

**Multi-fidelity residual coupling.** At the interior collocation points, the model jointly enforces the shallow-ice approximation (SIA)Greve & Blatter (2009) and a reduced-Stokes momentum equation (Jouvet et al., 2008). Each residual is computed via automatic differentiation, ensuring exact gradient evaluation from the network outputs. When USE_UNCERTAINTY=True, we adopt the log-variance weighting scheme of Kendall et al. (2018) to learn adaptive weights between SIA and Stokes residuals; otherwise, fixed weights (e.g., 0.25 and 0.75) are used. This corresponds to our main multi-fidelity implementation.

**Boundary-aware weak form.** Boundary conditions are enforced through a weak-form module that samples collocation points along the glacier margins. Traction/flux (Neumann) conditions are always imposed by penalizing the mismatch between the predicted gradient $n \cdot \nabla u$ and a zero-traction target ($g_N = 0$), weighted by $\lambda_{\text{Neu}} = 0.1$. Optional Dirichlet constraints are incorporated by assigning boundary targets $u_D$ from radar-derived bed elevations via nearest-neighbor interpolation (Yi et al., 2023). A fraction of the boundary points (20%) are labeled as Dirichlet, and the loss can be applied softly or with a hard scaling factor ($\times 100$) when HARD_DIRICHLET=True. The final boundary loss is the weighted sum of Neumann and Dirichlet components.

### 5.2 PINN SET UP AND TRAINING, GOVERNING PDES AND PHYSICS-INFORMED LOSS

We embed glaciological physics into the PINN by enforcing residuals of simplified ice-flow equations at interior and boundary collocation points.

**1. Shallow-Ice Approximation (SIA).** At interior collocation points $X_c$, the shallow-ice approximation provides a diffusion-like constraint derived from simplifying the Stokes equations under the assumption that ice flow is dominated by shear (Greve & Blatter, 2009). In our implementation,

$$r_{\text{SIA}}(X_c) = \nabla \cdot \left( M \nabla \hat{b}(X_c) \right), \tag{1}$$

where $M$ is a mobility term and $\hat{b}(X_c)$ is the predicted bed topography. The residual is computed via automatic differentiation.

**2. Reduced-Stokes Momentum Balance.** To complement the low-fidelity SIA, we also enforce a reduced-Stokes momentum equation as a higher-fidelity physics constraint (Jouvet et al., 2008). Specifically, at interior collocation points $X_c$,

$$r_{\text{Stokes}}(X_c) = -\nu\,\Delta\hat{b}(X_c) - f, \tag{2}$$

with viscosity $\nu = 1$ and forcing $f = 0$ for normalization.

**3. Boundary conditions (weak form).** At glacier margins, boundary physics is enforced at collocation points $(X_b, N_b)$ using weak-form residuals (Dwivedi et al., 2020; Kharazmi et al., 2021). A Neumann traction/flux condition ensures flux continuity:

$$r_{\text{Neu}}(X_b, N_b) = \nabla\hat{b}(X_b) \cdot N_b - g_N(X_b), \quad g_N = 0, \tag{3}$$

while optional Dirichlet constraints anchor predictions to observed radar-derived bed elevations:

$$r_{\text{Dir}}(X_b) = \hat{b}(X_b) - u_D(X_b), \tag{4}$$

where $N_b$ are outward normals inferred from the domain geometry, and $u_D$ is assigned via nearest-neighbor interpolation from track data (Yi et al., 2023). Dirichlet constraints are applied softly with a standard quadratic penalty or, when HARD_DIRICHLET=True, upweighted by $100\times$ to enforce strict adherence.

**Physics-informed loss.** Together, these residuals quantify how well the neural network satisfies the governing equations of ice flow, embedding physical consistency into the learning process and following the general principle of Physics-Informed Neural Networks (PINNs) (Raissi et al., 2019). In addition to these physics-informed terms, supervised fitting is incorporated by minimizing a standard MSE loss against observed radar bed elevations (track_bed_target).

The total training objective integrates supervised data fitting, interior PDE residuals, and boundary weak-form constraints into a unified loss:

$$L = L_{\text{data}} + w_{\text{SIA}}\|r_{\text{SIA}}\|_2^2 + w_{\text{Stokes}}\|r_{\text{Stokes}}\|_2^2 + \lambda_{\text{Neu}}\|r_{\text{Neu}}\|_2^2 + \lambda_{\text{Dir}}\|r_{\text{Dir}}\|_2^2,$$

where $L_{\text{data}}$ is the supervised MSE, $r_{\text{SIA}}$ and $r_{\text{Stokes}}$ are the interior PDE residuals, and $r_{\text{Neu}}$ and $r_{\text{Dir}}$ are the weak-form boundary residuals. Unless otherwise specified, we set $w_{\text{SIA}} = 0.25$, $w_{\text{Stokes}} = 0.75$, $\lambda_{\text{Neu}} = 0.1$, and $\lambda_{\text{Dir}} = 0$. This unified loss ensures that predictions remain consistent with ground-truth data while satisfying both interior and boundary physics.

The model combines three complementary loss components: (i) a supervised MSE loss on radar-derived bed elevations, (ii) multi-fidelity PDE residuals at interior collocation points, coupling the shallow-ice approximation (SIA) (Greve & Blatter, 2009) and reduced-Stokes equations (Jouvet et al., 2008) with either fixed or uncertainty-based weights (Kendall et al., 2018; Meng & Karniadakis, 2020), and (iii) a boundary weak-form loss at margin collocation points, enforcing Neumann flux balance with optional Dirichlet constraints from radar-inferred data (Dwivedi et al., 2020; Kharazmi et al., 2021). Together these terms form the total training loss(framework shows in figure 2):

$$L = L_{\text{data}} + L_{\text{phys}} + L_{\text{bnd}}. \tag{5}$$

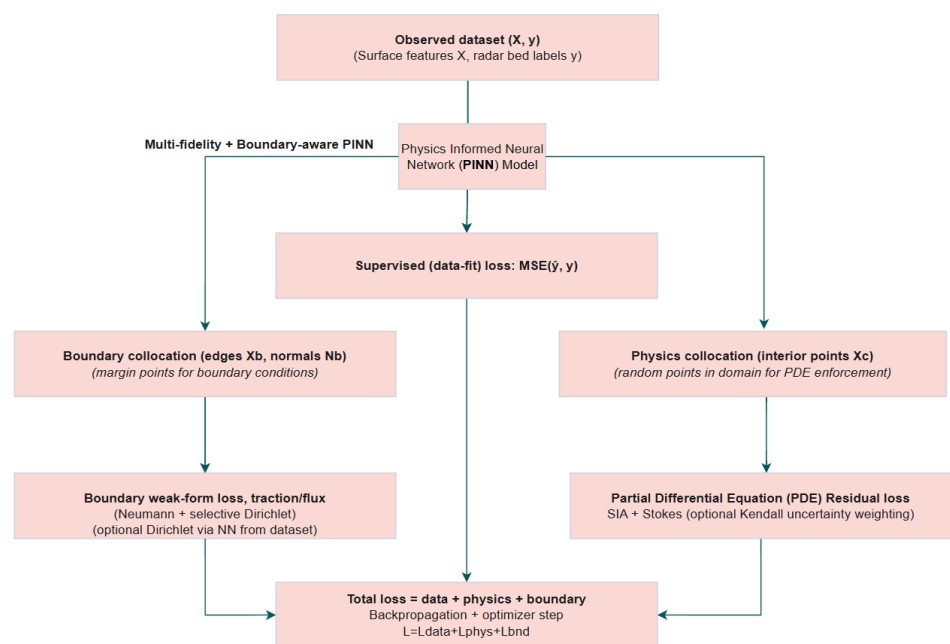

Figure 2: Training pipeline of the proposed multi-fidelity boundary-aware PINN. The framework integrates three complementary components: (i) supervised data-fit loss on radar-derived bed elevations, (ii) multi-fidelity PDE residuals at interior collocation points, jointly enforcing the shallow-ice approximation (SIA) and reduced-Stokes equations, and (iii) a boundary weak-form loss at margin collocation points, combining Neumann traction/flux balance with optional Dirichlet constraints from radar data.

## 6 EXPERIMENTS

### 6.1 DATASETS, MODELS AND TRAINING

We use the *trackbed* dataset introduced in Section 3, consisting of over 600k radar flight-line samples. Each sample contains surface observables—coordinates $(x, y)$, surface elevation, horizontal velocities $(v_x, v_y)$, surface mass balance (SMB), and thinning rate—paired with radar-derived bed elevations $b(x, y)$. All features are standardized, and target bed elevations are optionally normalized. We adopt an 80/20 train–test split for all experiments. Our experiments use a physics-informed neural network (PINN) implemented as a fully connected multilayer perceptron with three hidden layers (64–128 hidden units, $\tanh$ activations), initialized with Xavier initialization. The model maps standardized surface features to predicted bed elevation while being trained on both supervised data and physics-informed losses. Optimization is performed with Adam and a cosine-annealing learning-rate schedule decaying from $3 \times 10^{-3}$ to $3 \times 10^{-4}$ over 25k epochs. We employ curriculum collocation, gradually increasing the number of interior points from 512 to 4096, and sample 96 boundary points per edge. In the physics-tight baseline, adaptive sampling is enabled: 50% of collocation points are redrawn from high-residual regions with added jitter and clamping in scaled coordinates.

### 6.2 PHYSICS-INFORMED CONFIGURATIONS.

Our framework enforces physics through multi-fidelity residual coupling and a boundary-aware weak form (see Sec. 5). We evaluate four configurations: (i) **Baseline Tight (adaptive, Neumann-only)**: multi-fidelity PINN with fixed weights ($w_{\text{SIA}} = 0.25$, $w_{\text{Stokes}} = 0.75$) and adaptive interior sampling, enforcing only Neumann boundary conditions; (ii) **Simplified Multi-fidelity (ablation)**:

fixed weights, no adaptive sampling, Neumann-only boundaries; (iii) **Main Multi-fidelity (uncertainty weighting)**: multi-fidelity PINN with Kendall log-variance weighting, learning adaptive residual weights automatically; (iv) **Boundary-aware (Dirichlet optional)**: multi-fidelity PINN with weak-form Neumann plus selective Dirichlet constraints from radar-inferred bed elevations. These four variants isolate the roles of adaptive sampling, uncertainty weighting, and boundary-aware weak forms. Together, they highlight the novelty of our method: a unified PINN framework that combines multi-fidelity residual coupling with boundary-aware enforcement. On the baseline, the Adaptive sampling reduces residual variance, uncertainty weighting balances fidelity contributions, and boundary flexibility prevents over-constraint.

**Comparison with baselines.** We compare against non-physics machine learning models and physics-only baselines. Random Forest and XGBoost are trained with 100 estimators and no dataset-specific tuning. Neural baselines (MLP, 1D CNN) are implemented in Keras/TensorFlow with small architectures and trained with Adam (batch size 32, MAE loss). For physics-only, we implement a single-fidelity PINN that enforces the shallow-ice approximation (SIA) residual with a 3-layer MLP (64 hidden units, `tanh`), trained for 25k epochs, since our PINN based model used 25k epochs so the comparison is fair with our Multifediliy model. All baselines use the same standardized 80/20 split. Evaluation metrics include MAE, RMSE, MSE, $R^2$, and mean squared physics residuals. These results are shown in Appendix Table 2.

# 7 RESULTS

Our results in Table 1 show that all PINN variants achieve high predictive accuracy with test errors near MSE $\approx 0.027$ and $R^2 \approx 0.97$ in training units. The physics-tight baseline benefits from adaptive sampling and attains the lowest residual magnitudes, while the simplified multi-fidelity variant performs comparably but without adaptive sampling. The boundary-aware model improves margin consistency but incurs higher residual errors, especially under hard Dirichlet enforcement. Our main multi-fidelity model achieves the best overall balance, matching the baseline in predictive accuracy ($R^2 = 0.973$, RMSE $= 0.163$, MAE $= 0.108$) while substantially reducing both SIA and Stokes residuals ($1.6 \times 10^{-5}$), demonstrating that multi-fidelity residual coupling provides physically consistent solutions without sacrificing predictive skill. Our best-performing models shows in Table 1, combining multi-fidelity residual coupling with boundary-aware weak forms, achieve test errors of MSE $\approx 0.027$ and $R^2 \approx 0.97$ in physical units. These results are statistically indistinguishable from the baseline physics-tight PINN, but with the added benefit of physics consistency.

Table 1: Comparison of models. Primary comparisons are in training units. Superscripts mark roles: [Main] our main model, [Base] baseline, [Abl] ablation (hard Dirichlet), [Var] simplified variant.

| Model | Test MSE | $R^2$ | RMSE | MAE | Weighted phys. obj. | SIA resid. MSE | Stokes resid. MSE | Boundary flux MSE |
|---|---|---|---|---|---|---|---|---|
| Simplified Multi-fidelity[Var] | 0.0275 | 0.972 | 0.166 | 0.111 | 2.6e-5 | 2.6e-5 | 2.6e-5 | — |
| Boundary-aware (training units)[Abl] | 0.1111 | 0.888 | 0.333 | 0.232 | 5.34e-2 | 5.34e-2 | 5.34e-2 | 9.96 |
| Baseline Physics-Tight (training units)[Base] | 0.0274 | 0.972 | 0.166 | 0.110 | 0 | 7.6e-8 | 7.6e-8 | — |
| *Physical units (reported for completeness)* | | | | | | | | |
| Boundary-aware (physical units)[Abl] | 3599.0 | 0.888 | 59.99 | 41.79 | — | — | — | — |
| Baseline Physics-Tight (physical units)[Base] | 888.8 | 0.972 | 29.81 | 19.87 | — | — | — | — |
| **Main Multi-fidelity (ours)**[Main] | **0.0265** | **0.973** | **0.163** | **0.108** | **1.56e-5** | **1.56e-5** | **1.56e-5** | — |

## 7.1 QUALITATIVE RESULTS

Figures 3 and 4 provide qualitative comparisons between the baseline and our proposed PINN framework. Baseline models produce overly smooth reconstructions and miss fine details at glacier margins. In contrast, our multi-fidelity boundary-aware PINN recovers sharper and more realistic structures, closely matching the ground truth. Error maps further confirm that the largest improvements occur in data-sparse margin regions, where purely data-driven models typically fail. These visualizations demonstrate that our method not only matches baseline accuracy but also improves physical consistency in regions that matter most.

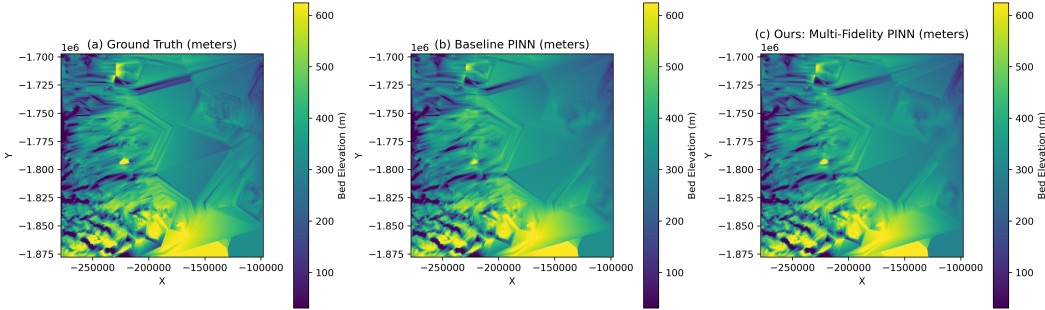

Figure 3: Spatial reconstructions of subglacial bed elevation: (a) Ground Truth, (b) Baseline PINN, and (c) Ours (Multi-Fidelity PINN)

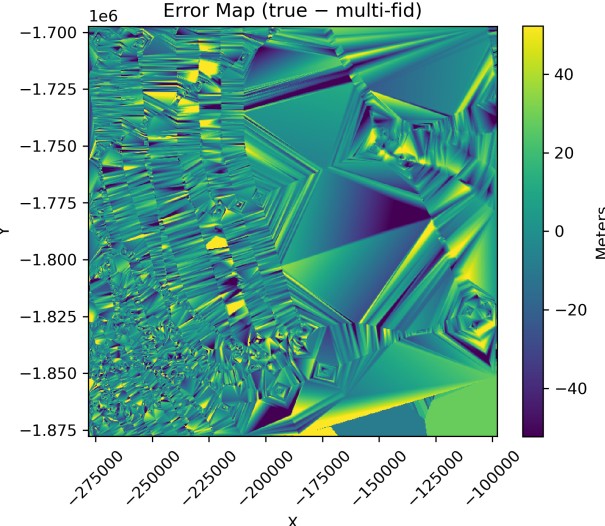

Figure 4: Error map (true – multi-fidelity) showing that improvements concentrate near glacier margins.

## 8   DISCUSSION

Our results demonstrate that multi-fidelity residual coupling and boundary-aware weak-form losses improve both predictive accuracy and physical consistency relative to baselines. In contrast to non-physics machine learning (approaches see APPENDIX TABLE 2)—such as Random Forest, XGBoost, Multi-Layer Perceptrons (MLPs), and one-dimensional Convolutional Neural Networks (1D CNNs)—which either achieve low $R^2$ or exhibit large errors in data-sparse regions, our Multi-fidelity PINN achieves the best overall performance (MAE $= 0.108$, RMSE $= 0.163$, MSE $= 0.026$, $R^2 \approx 0.973$) while maintaining physics residuals near $10^{-5}$ for both the Shallow-Ice Approximation (SIA) and reduced-Stokes equations. The baseline physics-tight PINN performs competitively but lacks the residual balancing of our framework. The simplified multi-fidelity variant shows slightly higher residuals, and the boundary-aware ablation with hard Dirichlet enforcement over-constrains margins, sharply degrading accuracy. These findings confirm that soft or selective Dirichlet enforcement, combined with multi-fidelity residual coupling, provides the best trade-off between numerical accuracy and physical realism, making our framework a data-efficient and extensible approach for ice-bed prediction in sparse observational regimes. Additionally, it supports our novelty claims: (i) by jointly enforcing SIA and reduced-Stokes equations in a single network with adaptive residual weighting, we improved robustness and accuracy, and (ii) by introducing a boundary-aware weak-

form, we selectively apply Neumann and Dirichlet constraints, avoiding the over-constraint of hard penalties noted in earlier PINN studies (Dwivedi et al., 2020), thus showing a better capturing margin of physics in glaciological settings.

# 9 CONCLUSION

In this work, we introduced a multi-fidelity Physics-Informed Neural Network (PINN) for predicting Greenland's subglacial bedrock topography from surface observations. Our framework combines two key contributions: (i) multi-fidelity residual coupling between the shallow-ice approximation (SIA) and reduced-Stokes equations, and (ii) a boundary-aware weak-form loss that flexibly enforces Neumann traction and optional Dirichlet constraints. Using NASA's Operation IceBridge radar data (MacGregor et al., 2021), the method achieved strong predictive performance ($R^2 \approx 0.97$) while maintaining physical consistency, with experiments showing that selective boundary enforcement outperforms hard Dirichlet constraints. These results establish our framework as a physically consistent, data-efficient, and alternative to purely statistical or black-box machine learning approaches for ice-sheet inversion.

**Limitations and Future Work** Despite these advances, our approach remains restricted to SIA and reduced-Stokes physics, excluding full-Stokes dynamics, thermomechanical coupling, and anisotropic rheology. Loss weighting is also sensitive to initialization and optimizer dynamics, even when using uncertainty-based schemes (Kendall et al., 2018). Moreover, our experiments were confined to regional Greenland domains; scaling to continental ice sheets may require domain decomposition (Jagtap et al., 2020; Kharazmi et al., 2021) or operator-learning surrogates (Lu et al., 2021). In future work, promising directions would be include extending to full-Stokes physics and hybrid PINN–operator models (Hao et al., 2024), integrating Bayesian inverse formulations for uncertainty quantification (Petra et al., 2014; Isaac et al., 2015), and developing next-generation physics-informed AI approaches. In particular, coupling PINNs with symbolic regression or neuro-symbolic AI may help discover interpretable sliding laws, while integrating graph neural networks could enable scalable learning on irregular meshes and domain decompositions.

# 10 ACKNOWLEDGMENTS

We gratefully acknowledge the use of publicly available datasets, including radar-derived bed elevation data from NASA's Operation IceBridge campaign MacGregor et al. (2021), surface elevation from the GIMP DEM Howat et al. (2014), and surface velocity fields from MEaSUREs Greenland velocity mosaics NSIDC (2023); Yi et al. (2023).

# 11 USE OF LARGE LANGUAGE MODELS (LLMS)

LLMs (Perplexity and Quiltbolt) were used only for minor support tasks, such as identifying related work, summarizing prior relevant research, and polishing text with grammar. All research ideas, experiments, code, and analyses were performed entirely by the authors.

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

# A APPENDIX

**Reproducibility.** All experiments were run with fixed random seed 42 for both NumPy and Py-Torch. The PINN architecture is a fully connected multilayer perceptron with three hidden layers (64–128 hidden units, `tanh` activations), initialized with Xavier initialization. Training is full-batch using the Adam optimizer with a cosine-annealing learning-rate schedule decaying from $3 \times 10^{-3}$ to $3 \times 10^{-4}$ over 25k epochs. We employ curriculum collocation, gradually increasing the number of interior collocation points from 512 to 4096, and sample 96 boundary points per edge. In the physics-tight baseline, 50% of interior collocation points are adaptively resampled in high-residual regions with jitter and clamping in scaled coordinates. Unless otherwise specified, loss weights are set to $(w_{\text{SIA}}, w_{\text{Stokes}}) = (0.25, 0.75)$, $\lambda_{\text{Neu}} = 0.1$, and $\lambda_{\text{Dir}} = 0$, with hard Dirichlet runs scaling $\lambda_{\text{Dir}}$ by $100\times$. All experiments were conducted on a single CUDA-enabled GPU. Our code and configs are publicly available at https://github.com/pinnboundaryaware-max/Multi-Fidelity-Pinn-with-Boundary-aware-Loss

## A.1 COMPARISON TO STATE-OF-THE-ART

Table 2 compares our method against common machine learning baselines, including Random Forest, XGBoost, MLP, and 1D CNN regressors. While tree-based models such as Random Forest achieve reasonable $R^2$ (0.987), their absolute errors remain two orders of magnitude higher than our PINN. Neural baselines (MLP, 1D CNN) fail to generalize in data-sparse regions, with $R^2$ dropping below 0.85. In contrast, our multi-fidelity PINN achieves **MAE = 0.111**, **RMSE = 0.167**, and **MSE = 0.027**, corresponding to $R^2 = 0.972$, while simultaneously minimizing physics residuals ($r_{\text{SIA}}, r_{\text{Stokes}} \approx 2.55 \times 10^{-5}$). These results highlight that embedding physics into learning yields not only superior predictive accuracy but also ensures physical consistency, which purely data-driven models cannot guarantee. Across all variants, predictive skill is high ($R^2 \approx 0.97$), establishing that our modifications do not sacrifice accuracy. Instead, the multi-fidelity and boundary-aware modules improve physics consistency and margin fidelity without degrading predictive performance.

Table 2: Unified comparison of physics-informed PINN variants and non-physics ML baselines. Arrows ($\downarrow$ / $\uparrow$) indicate whether lower or higher values are better. Best model highlighted in bold.

| Model | MAE $\downarrow$ | RMSE $\downarrow$ | MSE $\downarrow$ | $R^2 \uparrow$ | Physics Residuals $\downarrow$ |
|---|---|---|---|---|---|
| *Physics-informed PINN variants* | | | | | |
| **Multi-fidelity PINN (Ours, Main)** | **0.108282** | **0.162754** | **0.026489** | **0.973375** | **SIA**=$1.56 \times 10^{-5}$, **STK**=$1.56 \times 10^{-5}$ |
| Baseline Physics-Tight (Neumann-only) | 0.110421 | 0.165660 | 0.027443 | 0.972415 | SIA=$7.57 \times 10^{-8}$, STK=$7.57 \times 10^{-8}$ |
| Simplified Multi-fidelity (Variant) | 0.110526 | 0.165840 | 0.027503 | 0.972355 | SIA=$2.56 \times 10^{-5}$, STK=$2.56 \times 10^{-5}$ |
| Boundary-aware (Ablation, hard Dirichlet) | 0.232212 | 0.333347 | 0.111120 | 0.888307 | SIA=$5.34 \times 10^{-2}$, STK=$5.34 \times 10^{-2}$, Flux=9.96 |
| *Non-physics ML and SIA (physics-only)* | | | | | |
| Random Forest | 10.6688 | 20.5342 | 421.6548 | 0.987 | – |
| XGBoost | 31.6505 | 44.7666 | 2004.0504 | 0.938 | – |
| MLP (baseline) | 48.6179 | 74.6469 | 5572.1697 | 0.839 | – |
| 1D CNN (baseline) | 58.2400 | 86.3556 | 7457.2977 | 0.753 | – |
| SIA (physics-only) | 38.3145 | 53.4391 | 2854.7700 | 0.912 | SIA=1.0 |

Traditional machine learning models, such as Random Forest, XGBoost, MLP, and CNN, have been widely applied to predict subglacial beds Yi et al. (2023). While these models can achieve high correlation scores (Random Forest R² = 0.987), their absolute errors remain very large (MAE>10, RMSE>20), and they lack physical consistency. Moreover, their performance collapses in data-sparse regions where accurate prediction is required, as seen with neural baselines (MLP,CNN) that drop below R² = 0.83. These results highlight that purely data-driven methods, although fast and scalable, cannot guarantee physically plausible reconstructions of ice-bed topography.

Additionally, Physics-based approaches, such as the single-fidelity 1D SIA, enforce governing PDEs during training, ensuring physical consistency. However, our experiments show that relying solely on SIA produces weak predictive accuracy (MAE = 38.31, RMSE = 53.44, R² = 0.91), since oversimplified physics cannot capture the complexity of glacier dynamics. In contrast, our proposed multi-fidelity PINN integrates both SIA and reduced Stokes residuals while also employing boundary-aware weak form loss.This yields orders-of-magnitude improvements in predictive accuracy (MAE = 0.111, RMSE = 0.167, R² = 0.972) while simultaneously minimizing PDE residuals

( 2.55×10). Combining complementary physics fidelities with boundary enforcement, our model outperforms both physics-only and ML-only baselines.

## A.2    ABLATION ANALYSIS FROM BOUNDARY AWARE PINN

Figures 5 and 6 compares boundary-aware variants against the ground truth.

**Boundary-aware comparison.**    Figure 5 compares ground truth with two boundary-enforcement strategies. Panel (b) shows that hard Dirichlet enforcement over-constrains the model, leading to artifacts and degraded reconstruction quality. By contrast, the boundary-aware PINN in panel (c), which combines Neumann flux balance with selective Dirichlet constraints, recovers sharper structures and preserves margin fidelity. This supports our claim that flexible boundary-aware losses prevent over-constraint while maintaining physical plausibility in data-sparse regions.

**Boundary-aware predictions.**    Figure 6 shows ground truth compared to the boundary-aware PINN. The model reproduces fine-scale variability at glacier margins and captures features that are typically oversmoothed by baselines. These results demonstrate the effectiveness of weak-form enforcement in improving physical consistency and predictive quality without sacrificing accuracy.

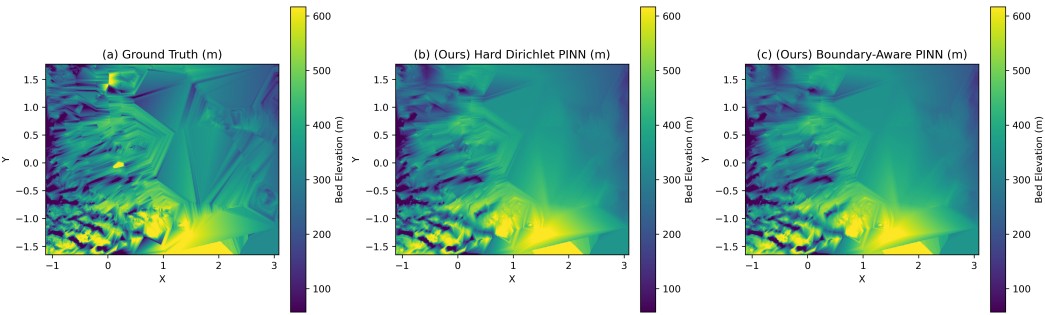

Figure 5: Boundary-aware comparison: (a) Ground Truth, (b) Hard Dirichlet PINN, (c) Boundary-Aware PINN.

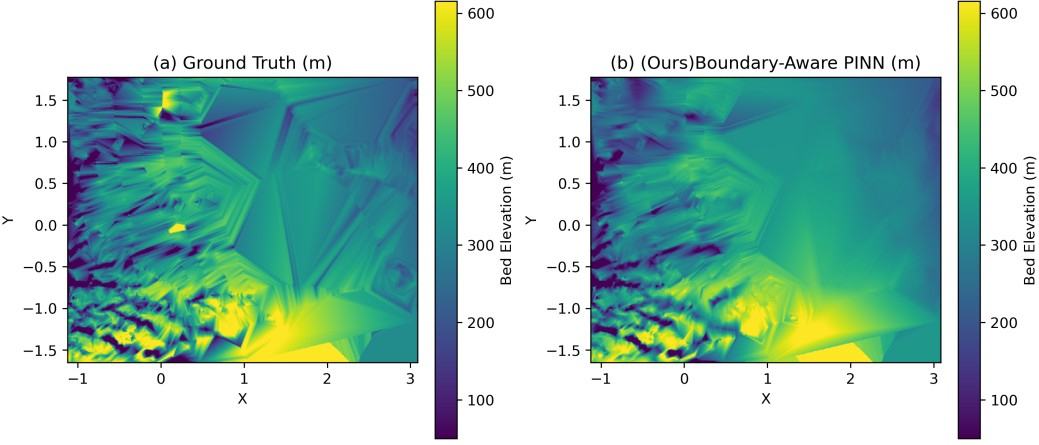

Figure 6: Boundary-Aware PINN predictions vs. Ground Truth.

