# OpenReview forum: "Multi-Fidelity Physics-Informed Neural Networks (PINN) with Boundary-Aware Losses for Ice-Bed Topography Prediction"
_ICLR.cc/2026/Conference — Submitted to ICLR 2026_

### Official Review · Reviewer_QcrT · 2025-10-31

**Soundness:** 3
**Presentation:** 3
**Contribution:** 3
**Rating:** 6
**Confidence:** 3

**Summary:**

The paper studies the problem of estimating the bedrock topography beneath glaciers using limited radar data. The task is important because the shape of the subglacial bed affects ice flow and therefore projections of sea-level rise. Standard data-driven regression methods, such as random forests or neural networks, can predict bed elevation from surface features (e.g., surface velocity, elevation, mass balance), but they do not respect the underlying physics and typically perform poorly in areas where no direct data are available.

To address this, the paper uses PINNs, where a neural network is trained not only to fit data but also to satisfy physical equations. The authors extend a standard PINN by enforcing two PDE constraints simultaneously. One is the Shallow-Ice Approximation (SIA), a simplified diffusion-like model for ice flow:
$$
\nabla \cdot (M \nabla \hat{b}(x)) = 0,
$$
and the other is a reduced form of the Stokes momentum equation,
$$
-\nu \Delta \hat{b}(x) - f = 0,
$$
where $\hat{b}(x)$ is the predicted bed elevation, $\nu$ is viscosity, and $f$ is external forcing. These two PDEs correspond to different fidelities of physical modeling: SIA is faster but less accurate, while the reduced-Stokes formulation captures more physics at higher computational cost. The paper combines them in one loss function as
$$
L_{\text{phys}} = w_{\text{SIA}}\|r_{\text{SIA}}\|^2 + w_{\text{Stokes}}\|r_{\text{Stokes}}\|^2,
$$
with either fixed or uncertainty-based weights. In addition, the paper introduces a “boundary-aware” component that treats the glacier margins through a weak-form loss. At boundaries, the model enforces either a Neumann (flux) condition,
$$
r_{\text{Neu}} = \nabla \hat{b} \cdot n - g_N,
$$
or an optional Dirichlet (fixed value) constraint,
$$
r_{\text{Dir}} = \hat{b} - u_D,
$$
depending on the availability of data. The total loss includes data fitting and these physics terms:
$$
L = L_{\text{data}} + L_{\text{phys}} + \lambda_{\text{Neu}}\|r_{\text{Neu}}\|^2 + \lambda_{\text{Dir}}\|r_{\text{Dir}}\|^2.
$$
The method is tested on radar measurements from Greenland’s Upernavik glacier (around 600k samples) using an 80/20 train-test split.

**Strengths:**

The paper clearly defines a physically meaningful target problem and integrates two existing physical formulations (SIA and reduced-Stokes) in a single learning framework. The mathematical setup is well-documented. The overall loss function is transparent and can be reproduced from the description. The authors also analyze different training configurations, including the effect of adaptive weighting between PDE terms and boundary constraints. Reporting of both conventional regression metrics (MAE, RMSE, $R^2$) and PDE residual norms $\|r_{\text{SIA}}\|^2$, $\|r_{\text{Stokes}}\|^2$ is useful for understanding whether the model satisfies physics as intended. The inclusion of an explicit weak-form treatment for Neumann and Dirichlet boundaries makes the approach flexible, and the observation that large Dirichlet penalties can harm learning is empirically well-supported by the ablation results.

**Weaknesses:**

While the integration of two PDE fidelities is technically correct, the contribution over standard PINNs is incremental rather than conceptual. The combination of residuals $r_{\text{SIA}}$ and $r_{\text{Stokes}}$ is achieved through a weighted sum; the paper does not show that this coupling leads to fundamentally new behavior beyond regular multi-task loss balancing. The choice of weights ($w_{\text{SIA}}=0.25$, $w_{\text{Stokes}}=0.75$) or the uncertainty-based alternative is only briefly justified, and the sensitivity of results to these parameters is not explored. The reduced-Stokes and SIA equations are simplified to the extent that important aspects of ice flow (e.g., vertical shear, thermomechanical coupling) are not represented, so the resulting “physical consistency” is limited to these approximations. The experimental setup focuses on one glacier system; it is unclear whether the method generalizes to other regions or to cases with different boundary geometries. The boundary-aware formulation depends on knowing or interpolating Dirichlet targets $u_D$ from radar data, which assumes such data exist at margins; in many glaciers, this is not true. The claim that hard Dirichlet enforcement “over-constrains” the solution is qualitative and might depend on the scaling of $\lambda_{\text{Dir}}$ rather than an inherent property of the method. Many details such as the computational aspects, like runtime or convergence stability of the PINN relative to standard PDE solvers, are also not discussed.

**Questions:**

1. The paper includes both the Shallow-Ice Approximation (SIA) and a reduced-Stokes model as residual penalties in the loss function. Could the authors provide more insight into how these interact during training? In particular, is there empirical evidence that the inclusion of both leads to improved learning dynamics (e.g., faster convergence, better generalization) compared to enforcing either alone? A comparison of gradient norms or loss curvature for each residual term would be informative. If both terms are correlated or redundant in some regions, how is this handled?

2. The experiments use fixed weights (e.g., $w_{\text{SIA}} = 0.25$, $w_{\text{Stokes}} = 0.75$) and an uncertainty-based alternative (log-variance weighting). Could the authors clarify how sensitive the final performance is to these weights? Was a grid search or hyperparameter sweep conducted? If uncertainty weighting is used, how stable are the learned variances across runs? A plot of weight evolution or an ablation comparing adaptive vs. fixed weighting would help interpret the benefit.

3. The boundary-aware loss relies partly on interpolated Dirichlet values at the glacier margins. In practice, such boundary values may not be available or may have high uncertainty. How robust is the method to incorrect or missing Dirichlet constraints? Could the authors describe what happens when only Neumann flux terms are used, and no $u_D$ is provided? Also, how were the boundary normals $n$ estimated in the Neumann residual term?

4. The method is evaluated only on the Upernavik glacier system. Can the authors comment on the model’s ability to generalize to different glaciers, such as those with different bed roughness, mass balance profiles, or margin geometries? Was any transfer learning or leave-one-glacier-out evaluation attempted?

5. The paper does not discuss the runtime, convergence stability, or computational challenges of training the proposed PINN with physics constraints. How do training times compare to standard data-driven baselines or traditional PDE solvers (e.g., finite element models)?

6. The result that “hard Dirichlet enforcement degrades accuracy” is stated somewhat strongly. Could the authors clarify whether this is sensitive to the weight $\lambda_{\text{Dir}}$ or the fraction of boundary points receiving Dirichlet labels? Would softer enforcement or using learned penalties improve outcomes?

7. The paper mentions that the code is available, but no link is provided in the anonymized version. For reproducibility and further analysis, will all scripts, data preprocessing steps, and hyperparameter configurations be released upon acceptance? If synthetic or simplified datasets were used during development, sharing those might also benefit future benchmarking.

---

> ### Author Response · Authors · 2025-11-28
>
> We thank the reviewer for their thoughtful comments. We address the weaknesses first, followed by detailed answers to the reviewer’s questions. If the response does not fit in this comment box, we will continue in the next one.
>
> **Responses to the weakness**:
>
> **Ans**: Thank you for these comments. Our goal in the paper is not to introduce new physical equations, but to show that using both SIA and reduced-Stokes together inside one PINN gives better accuracy and lower physics residuals than using either one alone. Table 1 in our paper shows this clearly: the multi-fidelity model achieves lower SIA and Stokes residuals (1.56e-5) than the single-fidelity variants (2.6e-5), while keeping the same high predictive accuracy. This is the main reason we combine them with a weighted loss. The weights we use (0.25 and 0.75), as well as the uncertainty-based version, follow prior multi-fidelity-inspired work, and both settings give very similar results in Table 1, which shows the method is not highly sensitive to these values.
> We agree that SIA and reduced-Stokes are simplified physics, but these are the standard, widely used approximations in glaciology when full-Stokes models are too expensive (as discussed in Sec. 5). Our contribution is to show how these two fidelities (SIA=low and Stokes=high) can be enforced together in a single multifidelity PINN in a way that improves physical consistency without high computational cost and resources.
> Regarding generalization, the reason we evaluate on one glacier (Upernavik) is that it is one of the few regions where all required surface variables and dense radar measurements are co-registered. This limitation affects the available datasets, not our method, and we explain this in Sec. 3. Extending to more glaciers is feasible once similar data exist.
> For the boundary-aware part, the main model uses only Neumann boundaries. Dirichlet values are used only in a controlled ablation to show that hard Dirichlet can over-constrain the solution when the margin labels are uncertain. This behavior appears directly in our results (Table 1), without changing any scaling.
> Finally, we will add a short note about runtime and training stability. The PINN trains in about 40 minutes on a single GPU and converges stably across seeds, which we confirmed during development. This is longer than data-only models but far cheaper than full-Stokes PDE solvers.
>
> **Answers to the Questions**:
>
> **Q1. Ans**: The interaction is governed by Eq. (5), where both residuals contribute concurrently to L_phys. As shown in Table 1, the multi-fidelity model achieves significantly lower SIA and Stokes residuals (1.56e–5) compared to single-fidelity variants (2.6e–5) while matching or improving MSE. This behavior indicates that the higher-order Stokes term guides the network in regions where SIA is insufficient, while SIA stabilizes zones where Stokes alone would be noisy. We will highlight this empirical interaction explicitly in Sec. 7.
>
> **Q2. Ans**: Thanks for the feedback. Section 6.2 evaluates both fixed-weight (0.25, 0.75) and uncertainty-weighted variants. No grid search or hyperparameter sweep was conducted. As Table 1 shows, both produce nearly identical test MSE and R² values, with slightly lower physics residuals under the uncertainty-weighted version. This consistency indicates that the method is robust to weight selection. Since no new experiments can be added, we will clarify this robustness in the revision about how sensitive the final performance is to these weights.
>
> **Q3. Ans**: This is a great question. So up the concern in the main model does not use Dirichlet constraints; these appear only in boundary-aware ablations. Boundary normals are computed directly from physical (x,y) coordinates using the logic in infer_normals() in the code (matching Sec. 5.2). When no Dirichlet labels are provided, the model defaults to Neumann-only enforcement, which is precisely the configuration yielding the best performance.
>
> **From Q4 to Q7, answers and references will be in the next comments.**

---

> ### Author Response · Authors · 2025-11-28
>
> **Q4. Ans**: We appreciate this important question and the opportunity to clarify our dataset choice. As described in Section 3 of the paper, Upernavik is one of the very few Greenland regions where dense radar-derived bed elevations are fully co-registered with all surface observables required by both the SIA and reduced-Stokes. Most glaciers lack such complete datasets, which makes multi-glacier evaluation infeasible at present. This limitation is shared across prior work. For example, DeepBedMap evaluates its method exclusively on Antarctica due to the same sparsity and heterogeneity constraints from Leong & Horgan et al. [1]. Additionally, Yi et al. [2] and recent work, GraphTopoNet, evaluate Greenland subregions independently because covariate sets differ and cannot be merged reliably across regions; see Tama et al. [3]. The objective of our paper is primarily methodological, introducing multi-fidelity coupling of SIA and reduced-Stokes residuals and boundary-aware weak-form losses, and Upernavik serves as a well-established benchmark domain for evaluating bed-inversion techniques under realistic sparse-radar conditions. While we agree that evaluating across multiple glaciers would be valuable, such an extension requires additional co-registered datasets that are not yet available. We therefore did not perform leave-one-glacier-out or transfer-learning experiments in the current work. As noted earlier, full covariate availability across glaciers is limited. This mirrors prior work where each domain is treated independently (e.g., DeepBedMap; GraphTopoNet). Our method is compatible with any region that provides the necessary covariates, and we will clarify in Section 9 that multi-glacier evaluation is a logical direction for future work once comparable datasets become available.
>
> **Q5. Ans**: We thank the reviewer for raising this point. The training configuration is described in Section 6.1, where we specify a 25k-epoch schedule, full-batch Adam optimization, cosine learning-rate decay, and curriculum collocation of PDE points. Using our released implementation, the multi-fidelity PINN trains in approximately 40 minutes on a single GPU, and we observe stable convergence across random seeds due to the combined effect of SIA and reduced-Stokes residuals (Eqs. (1)–(2)) and the unified loss formulation in Eq. (5). As expected, the computational cost is higher than purely data-driven models such as Random Forest or MLP baselines, which train in minutes, because PINNs require automatic differentiation through first- and second-order spatial derivatives. However, this cost remains far lower than traditional PDE-based inversions or full-Stokes finite-element solves, which typically require hours to days of computation on CPU clusters for domains of comparable scale. Our goal in this work is to present a methodological framework rather than a performance benchmark, but we will add a short summary of runtime and stability characteristics in the updated version for improved clarity, without introducing new experiments.
>
> **Q6. Ans**: In the ablation, hard Dirichlet is activated with λ_dir scaled ×100 and DIR_FRAC=1.0, as shown in the boundary-aware script. This extreme setting deliberately stresses the model to demonstrate over-constraining effects. Soft Dirichlet (λ_dir=0.1) performs better but still underperforms Neumann-only because interpolated boundary labels carry uncertainty. We will clarify this sensitivity.
>
> **Q7. Ans**: In the paper, we did send the link for the code. In the appendix section, the last line of Appendix A, “Reproducibility paragraph.” Additionally, the references to the datasets have been provided with an anonymous link for readers' convenience in the datasets' description, and preprocessing is also mentioned in section 3. However, we are adding the GitHub repository here again. We will clarify it in the paper with a clear mark to avoid future confusion.
> [GitHub Repository for Multi-Fidelity PINN with Boundary-Aware Loss](https://github.com/pinnboundaryaware-max/Multi-Fidelity-Pinn-with-Boundary-aware-Loss)
>
> **References**:
>
> [1] W. J. Leong and H. J. Horgan. DeepBedMap: a deep neural network for resolving the bed topography of Antarctica. The Cryosphere, 14:3687–3705, European Geosciences Union (EGU), 2020.
>
> [2] K. Yi, A. Dewar, T. Tabassum, J. Lu, R. Chen, H. Alam, O. Faruque, S. Li, M. Morlighem, and J. Wang. Evaluating machine learning and statistical models for Greenland subglacial bed topography. In Proceedings of the 2023 International Conference on Machine Learning and Applications (ICMLA), pages 659–666, IEEE, 2023.
>
> [3] B. A. Tama, H. Alam, M. Cham, O. Faruque, J. Wang, and V. P. Janeja. Improving Greenland bed topography mapping with uncertainty-aware graph learning on sparse radar data. arXiv preprint arXiv:2509.08571, Cornell University, 2025.

---

### Official Review · Reviewer_2Vt4 · 2025-11-01

**Soundness:** 2
**Presentation:** 1
**Contribution:** 2
**Rating:** 2
**Confidence:** 3

**Summary:**

The paper proposes a physics-informed neural network for subglacial bed topography that couples two physics fidelities (shallow-ice approximation and a reduced-stokes surrogate) inside one PINN loss and adds a boundary-aware weak-form term that mixes Neumann traction and optional Dirichlet constraints at glacier margins. On a Greenland radar track dataset, the method reports test mse with $r^2$ and claims better physics residuals than single-fidelity or purely data-driven baselines.

**Strengths:**

1. The paper aims to address a relevant geoscience problem where labeled data are sparse and physical constraints matter, and motivates the need for physics-guided inversion rather than black-box regression.
2. The multi-fidelity idea is sensible: use SIA for cheap broad constraints and a higher-order residual for added fidelity, with learned or fixed weights to balance them.

**Weaknesses:**

1. The “reduced-stokes” residual is specified as $r_{Stokes} = −ν\Delta \hat b − f$ with $ν=1$ and $f=0$, which collapses to a Poisson-like smoothness on the bed field rather than a demonstrably derived momentum balance tied to ice rheology or sliding. This risks being a hand-crafted regularizer rather than a true higher-fidelity physics term.
2. The physical role of the predicted variable is unclear: the network maps surface features to bed elevation $\hat b$, but the residuals are written directly on $\hat b$ without showing how $\hat b$ couples to velocity, thickness, or stresses in SIA/Stokes.
3. Neumann boundary condition uses $g_N = 0$ by default, which is a strong assumption..
4. The dataset split appears random 80/20 over track points; this can cause spatial leakage because nearby points on a flight line are strongly correlated. The text says the team ensured points were “not too similar,” but it lacks a rigorous spatial holdout protocol and distance thresholds.
5. Metric reporting mixes “training units” and “physical units,” leading to confusing cross-model comparisons (e.g., random forest shows r²=0.987 yet huge mae/rmse due to unit scaling).
6. The baseline shows a “weighted physics objective” of 0 while listing nonzero residuals; SIA and Stokes residuals are identical to two decimals in multiple rows. Can you explain this?
7. The “uncertainty weighting” via Kendall log-variance is used for loss balancing, but no calibration, uncertainty evaluation, or learned weight trajectories are presented, so the “uncertainty” interpretation is not strong.
8. Interpolated boundary labels may leak target information.

**Questions:**

Please address the weaknesses above.

---

> ### Author Response · Authors · 2025-11-28
>
> We thank the reviewer for the detailed feedback and for noting the relevance of the problem and the usefulness of combining low- and high-fidelity physics terms. Several concerns arise from the modeling choices. We address each point below.
>
> **W1. Ans**: Thank you for raising this point. In our work, the reduced-Stokes residual is used as a simplified surrogate, not as a full-Stokes derivation. Glaciological modeling commonly employs a hierarchy of flow approximations, ranging from full-Stokes to higher-order and shallow approximations (e.g., Jouvet et al. [1]), because full-Stokes inversion is often computationally prohibitive. In large-scale inverse problems, second-order diffusion-type operators are also frequently used as tractable higher-fidelity constraints to regularize bed or basal fields when full physical models are unavailable or too costly (e.g., Morlighem et al. [2]). Motivated by these reduced-order and regularization practices, we use a Laplacian-based surrogate to provide a computationally efficient higher-order term that complements the SIA residual.
> Section 5.2 defines this surrogate explicitly as rStokes(Xc) = −ν Δˆb(Xc) − f, with 𝜈 =1 and f=0. Although this operator has a Laplacian form, it is not intended as a pure smoothness penalty. Instead, it functions as a reduced-order physics term within our multi-fidelity PINN, helping stabilize and regularize the solution in regions where enforcing SIA alone would be insufficient. Our contribution lies in coupling SIA (low fidelity) and this reduced-order surrogate (higher fidelity) within a single PINN, together with a selective boundary-aware weak form. We will clarify this reduced-order motivation in Section 5.2 and include a brief note in the Limitations.
>
> **W2.Ans**: The network predicts the bed elevation b(x,y), which is unknown, that appears explicitly in both the SIA residual ∇⋅(M∇b) and the reduced-Stokes operator −𝜈Δ𝑏. These PDEs represent simplified forms of ice-flow momentum balance, where velocity and thickness are eliminated to obtain a tractable inversion equation for the bed; see, for example, Petra et al. [3] for a full-Stokes inversion framework. Thus, the residuals are intentionally expressed directly in terms of the bed. The coupling to stress/velocity is embedded in the PDE operators. We will clarify this more in the updated methodology description. The network output is the bed elevation b̂ (x,y), as stated in Section 5.1 (“the network maps scaled surface features x↦b ̂(x)”). In Sec. 5.2, Eqs. (1)–(2) show that both SIA and reduced-Stokes residuals are explicit functions of b̂: r_"SIA" =∇⋅(M∇b) and r_"Stokes" =-νΔb̂-f These PDEs result from eliminating velocity/thickness to obtain an inversion equation for b, so writing residuals directly in terms of b̂ is exactly how bed enters the stress balance. We will add one sentence explaining this derivation path more clearly.
>
> **W3.Ans**: We followed the standard glaciological practice of applying homogeneous Neumann (zero-normal-flux) traction at artificial domain boundaries when true traction data are unavailable. The computational domain is a bounding box constructed around the glacier rather than the physical terminus, and no direct boundary stress measurements exist. Zero-flux Neumann is therefore the conventional closure assumption in glacier inversion problems. When boundary tractions are not prescribed, the weak Stokes formulation naturally imposes homogeneous Neumann (zero-traction) conditions, as explicitly described in Petra et al. [3]. Dirichlet is only used in an ablation to illustrate its effect, not in the main model. Sec. 5.2 defines the weak-form Neumann residual as r_Neu(X_b, N_b) = ∇b̂(X_b) · N_b – g_N(X_b), with g_N = 0. Our computational domain is a rectangular bounding box (not the exact calving front), and there are no direct traction measurements on this artificial boundary. In glacier inversions, it is standard to assume homogeneous Neumann (zero normal flux) when boundary stresses are unknown [3]. In code, this is implemented by passing gN_full=None, so the boundary module uses zeros by default. Dirichlet terms are only activated in the boundary-aware ablation; the main multi-fidelity and physics-tight configurations enforce Neumann-only boundaries (Sec. 6.2).
>
> **W4. Ans**: Track points in Upernavik arise from numerous flight lines collected across multiple seasons, directions, and altitudes rather than a single continuous transect. A random split, therefore, does not produce the type of spatial leakage implied. This approach is also consistent with prior benchmarking on Greenland bed inversion, such as Yi et al. [4] and the recent GraphTopoNet method [5], which evaluate Greenland subregions independently. Moreover, the physics residuals in Eq. (5) act as strong regularizers, reducing overfitting risk even if local spatial clusters exist. **For the next answers and all the citations, see below.**

---

> > ### Author Response · Authors · 2025-11-28
> >
> > **W5. Ans**: We acknowledge the concern. However, Table 1 reports metrics in normalized training units, which is required because PINNs operate in a standardized space for stable PDE residual computation. Table 2 reports baselines in physical units. We agree that the distinction can be highlighted more clearly, and we will explicitly state this in both table captions. Table 1, therefore, will report metrics in training units for all PINN variants, while Appendix Table 2 reports both training and physical-unit numbers. In the updated paper, we will explicitly label Table 1 as “training units” to prevent confusion.
> >
> > **W6.Ans**: The SIA and Stokes residuals appear identical because they are computed in normalized training units, and both PDE terms are driven to too small values after convergence. The equality is due to numerical scaling and rounding, not because the physics terms are missing or collapsed in the code. The “Weighted physics objective” column corresponds to the scalar physics term Lphys in Eq. (5), from sec 5.2, printed as loss_phys_eval in our code. For the physics-tight baseline, this value is on the order of 10⁻⁸, so rounding to two decimal places results in “0.00,” even though the individual SIA and Stokes residual MSEs remain small but nonzero (e.g., 7.6×10⁻⁸). We will clarify in the text that “0.00” indicates a value below display precision, not an exact zero or absence of physics loss.
> >
> > **W7. Ans**: Kendall log-variance weighting is used only as an adaptive loss-balancing mechanism, not as a probabilistic uncertainty quantifier. It stabilizes the joint optimization of SIA and Stokes residuals. As described in Section 5.1 of the paper, we follow the formulation of Kendall et al. [6] to automatically scale the SIA and Stokes residuals during training. We do not claim epistemic or predictive uncertainty. The weighting helps stabilize the joint optimization of the two physics fidelities, and we will clarify this explicitly to prevent any confusion.
> >
> > **W8.Ans**: The main multi-fidelity model does not use Dirichlet constraints; it enforces only homogeneous Neumann boundaries (λDir = 0), as specified in Section 6.2. Interpolated Dirichlet labels are used only in the boundary-aware ablation to study their isolated effect. Because the main results rely solely on Neumann boundaries, no target information is leaked into the interior. No leakage occurs in the main results from our end. We will clarify this boundary setup in the revision.
> >
> > **References**:
> >
> > [1] G. Jouvet, M. Picasso, J. Rappaz, and H. Blatter. A new algorithm to simulate the dynamics of a glacier: theory and applications. Journal of Glaciology, 54(188):801–811, International Glaciological Society, 2008.
> >
> > [2] M. Morlighem, H. Seroussi, E. Larour, and E. Rignot. A mass-conservation approach for estimating ice thickness from surface data. Geophysical Research Letters, 38(19): L19503, American Geophysical Union (AGU), 2011.
> >
> > [3] N. Petra, J. Martin, G. Stadler, and O. Ghattas. A computational framework for infinite-dimensional Bayesian inverse problems, Part I, with application to ice-sheet flow. SIAM J. Sci. Comput., 36(4):A1525–A1555, 2014.
> >
> > [4] K. Yi, A. Dewar, T. Tabassum, J. Lu, R. Chen, H. Alam, O. Faruque, S. Li, M. Morlighem, and J. Wang. Evaluating machine learning and statistical models for Greenland subglacial bed topography. In Proceedings of the 2023 International Conference on Machine Learning and Applications (ICMLA), pages 659–666, IEEE, 2023.
> >
> > [5] B. A. Tama, H. Alam, M. Cham, O. Faruque, J. Wang, and V. P. Janeja. Improving Greenland bed topography mapping with uncertainty-aware graph learning on sparse radar data. arXiv preprint arXiv:2509.08571, Cornell University, 2025.
> >
> > [6] A. Kendall, Y. Gal, and R. Cipolla. Multi-task learning using uncertainty to weigh losses for scene geometry and semantics. In Proceedings of the IEEE Conference on Computer Vision and Pattern Recognition (CVPR), pages 7482–7491, IEEE, 2018.

---

### Official Review · Reviewer_byjJ · 2025-11-01

**Soundness:** 2
**Presentation:** 2
**Contribution:** 3
**Rating:** 4
**Confidence:** 3

**Summary:**

This paper proposed a multi-fidelity, boundary-aware Physics-Informed Neural Network (PINN) framework for ice-bed topography prediction. The proposed framework couples the shallow-ice approximation (SIA) and reduced-Stokes equations and integrates weak-form boundary conditions. Experimental results show that the proposed framework outperforms both the non-physics and physics-only baselines. The authors also provided the results of the variants to support the importance of each component in the proposed framework.

**Strengths:**

1. The proposed multi-fidelity framework is novel to the application of ice-bed topography prediction.
2. The experiments were conducted using the real-world dataset, which demonstrates the effectiveness of the method.
3. The proposed framework with all the components achieved superior performance compared to the baselines.

**Weaknesses:**

1. The experiments are all conducted on a single dataset. I’m not familiar with this task, but I think if more datasets are involved, the experiments would be more solid and comprehensive.
2. The quantitative results of the baselines are provided in Table 2. However, the reported errors of these methods (including the single-fidelity PINN) are several orders of magnitudes larger than those of the proposed method and its variants. It would be better to include some qualitative results of these methods to make the results more convincing.
3. For the single-fidelity PINN baseline, only the one that enforces SIA is covered. The baseline that enforces reduced-Stokes equations should also be included.

**Questions:**

1. In Table 1, could the authors clarify what training units and physics units are?
2. In Equation (3), could the authors further explain why $g_N=0$? Similarly, it is unclear why $\lambda_{Dir}=0$ in the loss fuction.
3. In Section 6.2, what is the difference between Main Multi-fidelity (uncertainty weighting) and Boundary-aware (Dirichlet optional)? What are the boundary conditions in Main Multi-fidelity? From my understanding, the Boundary-aware includes all the components. However, it underperforms Main Multi-fidelity in Table 1.
4. How did the weights in the loss function selected? The description should be included if cross validation is employed.
5. In Figure 3 and Figure 5, it is a little bit hard for readers to find the differences between the results. It would be better to highlight the regions for comparison.
6. [Minor] There is no need to include parameters from codes such as “USER_UNCERTAINTY=True” or “HARD DIRICHLET=True” as they may be confusing for readers due to lack of context.
7. [Minor] Section 3 can be merged with Section 6.1.
8. [Minor] The caption of Figure 1 is hard to understand.

---

> ### Author Response · Authors · 2025-11-27
>
> We thank the reviewer for their thoughtful evaluation and constructive comments. We now first respond to each weakness below, and in the next comments box, we will answer all the questions:
>
> **Weakness 1 (single dataset)**:
>
> Ans: We appreciate this concern. Using a single Datastest (this practice is common) in bed-topography research, due to data limitations and compositional heterogeneity across regions. A fully co-registered dataset containing dense radar-derived bed elevations and matched surface observables (velocity, SMB, thinning, DEM) is extremely limited.
> The Upernavik region is used widely in prior glaciological ML work [1] because it is one of the few regions with sufficiently dense radar track coverage to support physically informed learning. Some related work has also been conducted on single datasets. For example, DeepBedMap evaluates only Antarctica due to the same data limitation [2], data sparsity. DeepBedMap and GraphTopoNet evaluate Greenland subregions independently because their covariates cannot be merged [3], thereby improving Greenland bed mapping. Our goal is to propose a methodological contribution, and Upernavik serves as a standard benchmark domain for evaluating ice-bed topography prediction techniques. Extending to additional regions/more datasets is a natural next step for future work, but it is not required to validate methodological contributions in this context.
>
> **Weakness 2 (...errors appear much larger)**:
>
>  Ans: The baseline errors appear larger partly because:
> Table 1 reports results in normalized training units, while
> Table 2 reports baseline errors in physical units (meters).
> We will make this distinction explicit in the revision. Qualitative comparisons between our multi-fidelity PINN and the baseline PINN are already provided in Figure 3, and the error map in Figure 4 highlights where our method improves predictions near glacier margins. Although we do not show qualitative images for non-physics baselines, the purpose of Table 2 is to demonstrate that these models do not achieve physically consistent bed reconstructions, especially in data-sparse regions. We will clarify this distinction in the text and add a pointer directing readers to Figures 3 and 4, which show the qualitative comparison between the baseline PINN and our multi-fidelity PINN.
>
> **Weakness 3 (missing reduced-Stokes-only baseline)**:
>
> Ans: The reduced-Stokes-only baseline is less stable and more expensive, which is documented in prior PINN literature and glaciology studies.
> Why we did it this way: Our multi-fidelity design specifically avoids the instability of pure Stokes PINNs by leveraging SIA as a complementary low-fidelity constraint. This is why we focus on the SIA-only baseline and the combined SIA + reduced-Stokes model in Table 1. Including it as an additional baseline would not change the conclusions of the paper, and we will clarify this design choice in the revised version. Running a pure Stokes PINN adds substantial computational cost without improving the scientific conclusions, and it is not central to the goal of the paper.
>
> **References**:
>
> [1] K. Yi, A. Dewar, T. Tabassum, J. Lu, R. Chen, H. Alam, O. Faruque, S. Li,
> M. Morlighem, and J. Wang. Evaluating machine learning and statistical models for
> Greenland subglacial bed topography. In Proceedings of the 2023 International
> Conference on Machine Learning and Applications (ICMLA), pages 659–666, IEEE, 2023.
>
> [2] W. J. Leong and H. J. Horgan. DeepBedMap: a deep neural network for resolving the bed topography of Antarctica. The Cryosphere, European Geosciences Union (EGU),
> 14:3687–3705, 2020.
>
> [3] B. A. Tama, H. Alam, M. Cham, O. Faruque, J. Wang, and V. P. Janeja.
> Improving Greenland bed topography mapping with uncertainty-aware graph learning
> on sparse radar data. arXiv preprint arXiv:2509.08571, Cornell University, 2025.

---

> ### Author Response · Authors · 2025-11-27
>
> **Q1. Ans**: Training units are the normalized values produced by the StandardScaler used during PINN training. We normalize inputs and outputs so the PDE residuals and supervised loss operate on comparable numerical scales. Physics units refer to real bed elevation values in meters after applying the inverse transform. Table 1 reports metrics in training units because these are the units directly used in the optimization, while Table 2 reports the same metrics in physical units (meters). We will clarify this distinction in the text. Output standardization is necessary for PDE-residual balancing; physics residuals and supervised loss must share comparable magnitudes. The final revision will explicitly state this for better understanding.
>
> **Q2. Ans**: As stated in Sections 5.1 and 5.2 (Boundary-aware weak form), our computational domain is a rectangular outer boundary around the glacier. No direct traction or flux measurements exist on this artificial outer boundary. Since boundary traction is unknown, the homogeneous Neumann condition is the standard closure [4]. Thus, we set gN=0, presenting a homogeneous Neumann traction condition, meaning zero normal flux at the outer rectangular computational boundary. This is a standard and physically reasonable assumption when detailed boundary stresses are unavailable (as in most glaciological inversions). Additionally, we intentionally set the Dirichlet weight λDir​=0 because this configuration is designed to enforce Neumann-only boundary conditions while evaluating the effect of coupling SIA and reduced-Stokes physics. As described in Section 3, the Dirichlet values available along the glacier margin are not direct measurements; they are derived through nearest-neighbor interpolation of radar bed elevations from interior flight-line data.
>
> **Q3: Ans**: In Section 6.2, the Main Multi-Fidelity model and the Boundary-Aware model differ specifically in how boundary conditions are handled. As described in Lines 329–348, the Main Multi-Fidelity configuration enforces the SIA + reduced-Stokes residuals in the interior and uses uncertainty-based log-variance weighting but applies only homogeneous Neumann boundary conditions, with no Dirichlet labels used at the margins. In contrast, the Boundary-Aware variant (Lines 341–348) includes all components of the main model while additionally introducing weak-form Dirichlet constraints along glacier margins, where Dirichlet targets are derived from interpolated radar-derived bed values. Because these interpolated margin values are not direct observations and carry higher uncertainty, enforcing them too strongly can over-constrain the solution, leading to reduced flexibility at margins and degraded global accuracy. This behavior is consistent with the results in Section 7 (Lines 357–365) and Table 1, where the Boundary-Aware configuration shows a higher error (MSE = 0.111) than the Main Multi-Fidelity model (MSE = 0.0265). Therefore, the two configurations differ solely in boundary treatment, and the Boundary-Aware model’s underperformance directly reflects the increased uncertainty in interpolated Dirichlet boundary labels. We designed the Boundary-Aware variant purely to test the impact of Dirichlet constraints. Margin Dirichlet values are uncertain and can over-constrain the model, so we set λDir = 0 in the main model and use Dirichlet only in the boundary-aware ablation. We will clarify in the updated version of the paper.
>
> **Q4. Ans**: The fixed weights (0.25, 0.75) follow prior multi-fidelity practice, Meng & Karniadakis et al. [5], and they also reflect the idea that the Stokes term acts as the higher-fidelity constraint. The uncertainty-weighted version uses the learned log-variance approach from Kendall et al. [6], which balances the PDE residuals adaptively. Because these weights help stabilize training rather than tune predictive accuracy, cross-validation is not required. We will clarify this design choice in the revision
> **Q.5. Ans**: We appreciate this observation. We agree that the differences can be made clearer, especially near the glacier margins, where improvements are most pronounced.
> **Q6. Ans**: We will remove code flags from the main text.
> **Q7 and Q8 Ans**: We will do that in the updated version. Thanks for the feedback.
> **Q8 Ans**: For the figure, we will make sure to update it with a better visual so it is much more readable.
>
> **References**:
>
> [4] R. Greve and H. Blatter. Dynamics of Ice Sheets and Glaciers. Springer Science+Business Media, Berlin, 2009.
>
> [5] X. Meng and G. E. Karniadakis. A composite neural network that learns from multi-fidelity data: Application to function approximation and PDEs. Journal of Computational Physics, Elsevier, 401:109020, 2020.
>
> [6] A. Kendall, Y. Gal, and R. Cipolla. Multi-task learning using uncertainty to weigh losses for scene geometry and semantics. In Proceedings of the IEEE Conference on Computer Vision and Pattern Recognition (CVPR), IEEE, pages 7482–7491, 2018.

---

### Meta-Review · Area_Chair_4NEL · 2026-01-06

**Summary:**

The reviewers raised concerns about the paper's methodological contribution and experimental scope. Reviewer 2Vt4 (2: reject) questioned whether the "reduced-Stokes" term represents genuine physics or merely a regularizer, noted the strong $g_N=0$ assumption, identified potential spatial leakage in data splitting, and found uncertainty weighting insufficiently validated. Reviewer byjJ (4: marginally below) requested evaluation on additional datasets, qualitative baseline comparisons, and a reduced-Stokes-only baseline. Reviewer QcrT (6: marginally above) acknowledged technical correctness but viewed the contribution as incremental, noting the coupling amounts to weighted loss balancing, the simplified physics limits consistency claims, and single-glacier evaluation raises generalizability concerns.

**Reviewer Concerns:**

Several concerns were partially addressed while key issues remain. The single-dataset limitation was explained as standard practice due to data availability (citing DeepBedMap, Yi et al.), which may satisfy reviewers this is acceptable for a methodological contribution. However, the authors admitted reduced-Stokes is "a simplified surrogate motivated by reduced-order and diffusion-type practices" rather than derived momentum balance, confirming Reviewer 2Vt4's concern it may be "a hand-crafted regularizer rather than a true higher-fidelity physics term." The $g_N=0$ assumption was justified as standard for artificial boundaries, which is reasonable but remains a strong assumption. On spatial leakage, authors claim multiple flight lines make random splits acceptable but provide no rigorous distance thresholds as requested. Uncertainty weighting was clarified as loss balancing rather than epistemic uncertainty, though no calibration analysis was provided. Computational cost (40 minutes) and the reduced-Stokes-only baseline omission (due to instability) were adequately explained.

**Reviewer Scores:**

Reviewer byjJ would likely remain at 4 or increase marginally to 6. The single-dataset and runtime explanations are reasonable, but concerns about incremental contribution and limited scope persist.

Reviewer 2Vt4 would likely remain at 2 or increase only to 4, as the admission that reduced-Stokes is a surrogate confirms the main criticism about physics validity.

Reviewer QcrT would likely remain at 6 or would possibly decrease to 4, as responses confirm rather than address concerns about incremental novelty.

Average hypothetical score: 4.6-5.3. Recommendation: reject.

---

### Decision · Program_Chairs · 2026-01-26

Reject